# Non-Hydrostatic RegCM4 (RegCM4-NH): Model description and case studies over multiple domains.

Erika Coppola (1), Paolo Stocchi (2), Emanuela Pichelli (1), Jose Abraham Torres Alavez (1), Russel Glazer (1), Graziano Giuliani (1), Fabio Di Sante (1), Rita Nogherotto (1), Filippo Giorgi (1)

*Correspondence to*: Erika Coppola (coppolae@ictp.it)

1. International Centre for Theoretical Physics (ICTP),Trieste, Italy
2. Institute of Atmospheric Sciences and Climate, National Research Council of Italy, CNR-ISAC, Bologna, Italy

**Abstract.** We describe the development of a non-hydrostatic version of the regional climate model RegCM4, called RegCM4-NH, for use at convection-permitting resolutions. The non-hydrostatic dynamical core of the Mesoscale Model MM5 is introduced in the RegCM4, with some modifications to increase stability and applicability of the model to long-term climate simulations. Newly available explicit microphysics schemes are also described, and three case studies of intense convection events are carried out in order to illustrate the performance of the model. They are all run at convection-permitting grid spacing of 3 km over domains in northern California, Texas and the Lake Victoria region, without the use of parameterized cumulus convection. A substantial improvement is found in several aspects of the simulations compared to corresponding coarser resolution (12 km) runs completed with the hydrostatic version of the model employing parameterized convection. RegCM4-NH is currently being used in different projects for regional climate simulations at convection-permitting resolutions, and is intended to be a resource for users of the RegCM modeling system.

**Keywords**
Regional climate models; RegCM4; km-scale resolution; climate change

**Introduction**

Since the pioneering work of Dickinson et al. (1989) and Giorgi and Bates (1989), documenting the first regional climate modeling system (RegCM, version 1) in literature, the dynamical downscaling technique based on limited area Regional Climate Models (RCMs) has been widely used worldwide, and a number of RCM systems have been developed (Giorgi 2019). RegCM1 (Dickinson et al., 1989, Giorgi and Bates, 1989) was originally developed at the National Center for Atmospheric Research (NCAR) based on the Mesoscale Model version 4 (MM4) (Anthes et al, 1987) . Then, further model versions followed: RegCM2 (Giorgi et al. 1993a,b), RegCM2.5, (Giorgi and Mearns 1999), RegCM3 (Pal et al. 2007), and lastly RegCM4 (Giorgi et al 2012). Except for the transition from RegCM1 to RegCM2, in which the model dynamical core was updated from that of the MM4 to that of the MM5 (Grell et al. 1994), these model evolutions were mostly based on additions of new and more advanced physics packages. RegCM4 is today used by a large community for numerous projects and applications, from process studies to paleo and future climate projections, including participation in the Coordinated Regional Downscaling EXperiment (CORDEX, Giorgi et al. 2009; Gutowski et al. 2016). The model can also be coupled with ocean, land and chemistry/aerosol modules in a fully interactive way (Sitz et al. 2017).

The dynamical core of the standard version of RegCM4 is hydrostatic, with sigma-p vertical coordinates. As a result, the model can be effectively run for grid spacings of ~10 km or larger, for which the hydrostatic assumption is valid. However, the RCM community is rapidly moving to higher resolutions of a few km, i.e. "convection-permitting" (Prein et al. 2015; Coppola et al. 2020) and therefore the dynamical core of RegCM4 has been upgraded to include a non-hydrostatic dynamics representation usable for very high resolution applications. This upgrade, which we name RegCM4-NH, is essentially based on the implementation of the MM5 non-hydrostatic dynamical core within the RegCM4 framework, which has an entirely different set of sub-grid model physics compared to MM5.

RegCM4-NH is already being used in some international projects focusing on climate simulations at convection-permitting km-scales, namely the European Climate Prediction

System (EUCP, Hewitt and Lowe 2018) and the CORDEX Flagship Pilot Study dedicated
to convection (CORDEX-FPSCONV, Coppola et al. 2020), and it is starting to be used
more broadly by the RegCM modeling community.
For example, the recent papers by Ban et al. (2021) and Pichelli et al. (2021) document
results of the first multi-model experiment of 10-year simulations at the convection-
permitting scales over the so-called greater Alpine region. Two different simulations with
RegCM4-NH for present day conditions have contributed to the evaluation analysis of
Ban et al. (2021). They were carried out at the International Centre for Theoretical Physics
(ICTP) and the Croatian Meteorological and Hydrological Service (DHMZ) using two
different physics configurations. The results show that RegCM4-NH largely improves the
precipitation simulation as compared to available fine scale observations when going from
coarse to high resolution, in particular for higher order statistics, such as precipitation
extremes and hourly intensity. Pichelli et al. (2021) then  analyse multi-model ensemble
simulations driven by selected CMIP5 GCM projections for the decades 1996–2005 and
2090–2099 under the RCP8.5 scenario. ICTP contributed to the experiment with
simulations using RegCM4-NH driven by the MOCH-HadGEM GCM (r1i1p1) in a two
level nest configuration (respectively at 12 and 3 km grid). The paper shows new insights
into future changes, for example an enhancement of summer and autumn hourly rainfall
intensification compared to coarser resolution model experiments, as well as an increase
of frequency and intensity of high-impact weather events.

In this paper we describe the structure of RegCM4-NH and provide some illustrative
examples of its performance, so that model users can have a basic reference providing
them with background information on the model. In the next section we first describe the
new model dynamical core, while the illustrative applications are presented in section 4.
Section 5 finally provides some discussion of future developments planned for the RegCM
system.

**Model description**

**Model description**
In the development of RegCM4-NH, the RegCM4 as described by Giorgi et al. (2012) was
modified to include, the non-hydrostatic dynamical core (*idynamic* = 2 namelist option as
described in RegCM-4.7.1/Doc/README.namelist of the source code) of the mesoscale
model MM5 (Grell et al. 1994). This dynamical core was selected because RegCM4
already has  the same grid and variable structure as MM5 in its hydrostatic core, which
substantially facilitated its implementation (Elguindi et al. 2017).

The model equations with complete description of the Coriolis force and a top radiative
boundary condition, along with the finite differencing scheme, are given in Grell et al.
(1994). Pressure, p, temperature, T, and density, $\varrho$, are first decomposed into a
prescribed reference vertical profile plus a time varying perturbation. The prognostic
equations are then calculated using the pressure perturbation values. Compared to the
original MM5 dynamical core, the following modifications were implemented in order to
achieve increased stability for long term climate simulations (Elguindi et al. 2017
document any modifications which follow the choice of the non-hydrostatic dynamical
core through the namelist parameter *idynamic* = 2; further available user-dependant
options, and the corresponding section in the namelist, are explicitly indicated):

i) The reference state temperature profile is computed using a latitude dependent
climatological temperature distribution and thus is a function of the specific domain
coordinates (*base_state_pressure*, *logp_lrate* parameters in *&referenceatm*) (Elguindi et
al. 2017). These two parameters were hard-coded in the original MM5 while for the
RegCM are user configurable;

ii) The lateral time dependent boundary conditions (*iboudy* in *&physicsparam*) for each
prognostic variable use the same exponential relaxation technique (*iboudy* = 5) described
in Giorgi et al. (1993). The linear MM5 relaxation scheme is also kept as an option (*iboudy*
= 1);

iii) The advection term in the model equations, which in the MM5 code is implemented
using a centered finite difference approach, was changed to include a greater upstream
weight factor as a function of the local Courant number (Elguindi et al. 2017). The
maximum value of the weight factor is user configurable (*uoffc* in *&dynparam*). As detailed
in the MM5 model description (Grell et al., 1994), the horizontal advection term for a scalar
variable X contributes to the total tendency as:

$$\Delta_{adv}\left(p^*X\right)_G = -m^2|_G \left[ \frac{\left(p^*X|_b\frac{u}{m}|_b - p^*X|_a\frac{u}{m}|_a\right)}{dx} + \frac{\left(p^*X|_d\frac{v}{m}|_d - p^*X|_c\frac{v}{m}|_c\right)}{dy} \right]$$



where the $m$ is the projection mapping factor and, with respect to Figure 1, assuming that
the computation is to be performed for the gold cross point $G$, the averages are performed
in the points $a, b, c, d$. For the $u/m$ and $v/m$ terms, the average value is computed using
respectively the values in points $AC, BD, CD, AB$.
In RegCM4 for the term $p^*X$, the model computes a weighted average value of the field
using the value in gold+cyan and gold+green cross points with weights increasing the
relative contribution of the upstream point up as a function  of the local courant number:

$p^*X|_a = 0.5((1 - f_1)p^*X|_G + (1 + f_1)p^*X|_{c_1})$
$p^*X|_b = 0.5((1 - f_1)p^*X|_{c_2} + (1 + f_1)p^*X|_G)$
$p^*X|_c = 0.5((1 - f_2)p^*X|_G + (1 + f_2)p^*X|_{g_1})$
$p^*X|_d = 0.5((1 - f_2)p^*X|_{g_2} + (1 + f_2)p^*X|_G)$
where $f_1, f_2$ are defined as the local Courant number for the 1D advection equations
multiplied for a control factor:

$f_1 = \mu_{fc}dt\frac{(u|_a + u|_b)}{2dx}$
$f_2 = \mu_{fc}dt\frac{(v|_c + v|_d)}{2dy}$   ;


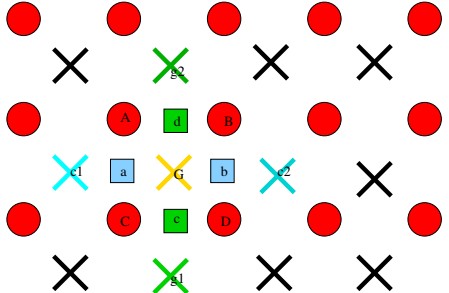


**Figure 1 Schematic representation showing the horizontal advection scheme staggering. Circles are U,V points. X are scalar variable points.**



iv) The water species (cloud, ice,rain, snow) term uses the same advection scheme as
the other variables (Elguindi et al. 2017) and not a complete upstream scheme as in the
MM5 code (Grell et al. 1994);

v) A local flux limiter reduces the advection terms in order to remove unrealistic strong
gradients and its limits are user configurable (in the *&dynparam* section the maximum
gradient fraction for advection: temperature, *t_extrema*, specific humidity, *q_rel_extrema*,
liquid cloud content, *c_rel_extrema* and for tracers, *t_rel_extrema*). This was hardcoded
in the MM5 code and the limits were not user configurable;

vi) The diffusion stencil of the Laplace equation uses a nine point approach as in LeVeque
(2006) and a topography dependent environmental diffusion coefficient is added to
reduce spurious diffusion along pressure coordinate slopes (Elguindi et al. 2017) as in
the hydrostatic version of the code (Giorgi et al. 1993b). The change in stencil does not
affect the overall fourth order precision of the model, but reduces the computational
stencil size, thus reducing the communication overhead;

vii) The top boundary radiative condition (*ifupr* = 1 in *&nonhydroparam*) adopted in the
semi-implicit vertical differencing scheme to reduce the reflection of energy waves uses
coefficients on a 13x13 matrix which are re-computed every simulation day and not kept
constant throughout the whole simulation as in the MM5 code. This allows the model to
be run for longer simulation times while not being strongly tied to the initial atmospheric
conditions;

viii) The dynamical control parameter β in the semi-implicit vertical differencing scheme
(*nhbet* in *&nonhydroparam*) used for acoustic wave damping (Elguindi et al. 2017) is user
configurable (Klemp and Dudhia, 2008), while it is hard-coded in the MM5;

ix) A Rayleigh damping (*ifrayd* = 1 in *&nonhydroparam*) of the status variables towards
the input GCM boundary conditions can be activated in the top layers (*rayndamp*
configuring the number of top levels to apply) with a configurable relaxation time
(*rayalpha0*, Klemp and Lilly, 1978, Durran and Klemp, 1983. This is consistent to what is
implemented in the WRF model);

x) The water species time filtering uses the Williams (2009) modified filter with α = 0.53
instead of the RA filter used by all the other variables. The ν factor in the RA filter is user
configurable (*gnu1* and *gnu2* in *&dynparam*). This reduces the damping introduced by the
Robert-Asselin filter and the computational diffusion introduced by the horizontal
advection scheme.

With these modifications, the model basic equations, under leap-frog integration scheme,
are (Elguindi et al. 2017) :


$$\frac{\partial p^* u}{\partial t} = -m^2 \left[ \frac{\partial p^* uu/m}{\partial x} + \frac{\partial p^* vu/m}{\partial y} \right] - \frac{\partial p^* u\dot\sigma}{\partial \sigma} + uDIV - \frac{mp^*}{\rho} \left[ \frac{\partial p'}{\partial x} - \frac{\sigma}{p^*} \frac{\partial p^*}{\partial x} \frac{\partial p'}{\partial \sigma} \right] + p^* fv - p^* ew \cos\theta + D_u \qquad (1)$$


$$\frac{\partial p^* v}{\partial t} = -m^2 \left[ \frac{\partial p^* uv/m}{\partial x} + \frac{\partial p^* vv/m}{\partial y} \right] - \frac{\partial p^* v\dot\sigma}{\partial \sigma} + vDIV - \frac{mp^*}{\rho} \left[ \frac{\partial p'}{\partial y} - \frac{\sigma}{p^*} \frac{\partial p^*}{\partial y} \frac{\partial p'}{\partial \sigma} \right] - p^* fu + p^* ew \sin\theta + D_v \qquad (2)$$


$$\frac{\partial p^* w}{\partial t} = -m^2 \left[ \frac{\partial p^* uw/m}{\partial x} + \frac{\partial p^* vw/m}{\partial y} \right] - \frac{\partial p^* w\dot\sigma}{\partial\sigma} + wDIV +$$

$$p^* g \frac{\rho_0}{\rho} \left[ \frac{1}{p^*} \frac{\partial p'}{\partial\sigma} + \frac{T_v'}{T} - \frac{T_0 p'}{T p_0} \right] - p^* g \left[ (q_c + q_r) \right] + p^* e \left( u\cos\theta - v\sin\theta \right) + D_w \quad (3)$$



$$\frac{\partial p^* p'}{\partial t} = -m^2 \left[ \frac{\partial p^* up'/m}{\partial x} + \frac{\partial p^* vp'/m}{\partial y} \right] - \frac{\partial p^* p'\dot\sigma}{\partial\sigma} + p'DIV -$$

$$m^2 p^* \gamma p \left[ \frac{\partial u/m}{\partial x} - \frac{\sigma}{mp^*} \frac{\partial p^*}{\partial x} \frac{\partial u}{\partial\sigma} + \frac{\partial v/m}{\partial y} - \frac{\sigma}{mp^*} \frac{\partial p^*}{\partial y} \frac{\partial v}{\partial\sigma} \right] + \rho_0 g\gamma p \frac{\partial w}{\partial\sigma} + p^* \rho_0 g \quad (4)$$



$$\frac{\partial p^* T}{\partial t} = -m^2 \left[ \frac{\partial p^* uT/m}{\partial x} + \frac{\partial p^* vT/m}{\partial y} \right] - \frac{\partial p^* T\dot\sigma}{\partial\sigma} + TDIV +$$

$$\frac{1}{\rho c_p} \left[ p^* \frac{Dp'}{Dt} - \rho_0 g p^* w - D_{p'} \right] + p^* \frac{\dot Q}{c_p} + D_T \quad\quad (5)$$



Where:

$$DIV = m^2 \left[ \frac{\partial p^* u/m}{\partial x} + \frac{\partial p^* v/m}{\partial y} \right] + \frac{\partial p^* \dot\sigma}{\partial\sigma}$$


$$\dot\sigma = -\frac{\rho_0 g}{p^*} w - \frac{m\sigma}{p^*} \frac{\partial p^*}{\partial x} u - \frac{m\sigma}{p^*} \frac{\partial p^*}{\partial y} v$$


$$\tan\theta = -\cos\phi \frac{\partial\lambda/\partial y}{\partial\phi/\partial x}$$


$$p(x,y,z,t) = p_0(z) + p'(x,y,z,t)$$
$$T(x,y,z,t) = T_0(z) + T'(x,y,z,t)$$
$$\rho(x,y,z,t) = \rho_0(z) + \rho'(x,y,z,t)$$

with the vertical sigma coordinate defined as:

$$\sigma = \frac{(p_0 - p_t)}{(p_s - p_t)}$$



$p_s$ is the surface pressure and $p_0$ is the reference pressure profile. The total pressure
at each grid point is thus given as:

$$p(x, y, z, t) = p^* \sigma(k) + p_t + p'(x, y, z, t)$$


With $p_t$ being the top model pressure assuming a fixed rigid lid.
The model physics schemes for boundary layer, radiative transfer, land and ocean
surface processes, cloud and precipitation processes are extensively described in Giorgi
et al. (2012) and summarized in Table 1. For each physics component a number of
parameterization options are available (Table 1), and can be selected using a switch
selected by the user. As mentioned, the use of non-hydrostatic dynamics is especially
important when going to convection-permitting resolutions of a few km (Prein et al. 2015).
At these resolutions the scale separation assumption underlying the use of cumulus
convection schemes is not valid any more, and explicit cloud microphysics
representations are necessary. The RegCM4 currently includes two newly implemented
microphysics schemes, the Nogherotto-Tompkins (Nogherotto et al. 2016) and the WSM5
scheme from the Weather Research Forecast (WRF, Skamarok et al. 2008) model, which
are briefly described in the next sections for information to model users.

| Model physics (*Namelist flag*) | Options | *n. option* | Reference |
|---|---|---|---|
| **Dynamical core** (*idynamic*) | Hydrostatic | 1 | Giorgi et al. 1993a,b Giorgi et al. 2012 |
| | Non-Hydrostatic (*) | 2 | present paper |
| **Radiation** (*irrtm*) | CCSM | 0 | Kiehl et al. 1996 |
| | RRTM (*) | 1 | Mlawer et al. 1997 |
| **Microphysics** | Subex | 1 | Pal et al 2000 |

| (*ipptls*) | Nogherotto Thompkins | 2 | Nogherotto et al. 2016 |
|---|---|---|---|
| | WSM5 (*) | 3 | Hong et al 2004 |
| **Cumulus** (*icup*) | Kuo | 1 | Anthes et al. 1987 |
| | Grell | 2 | Grell 1993 |
| | Emanuel | 4 | Emanuel 1991 |
| | Tiedtke | 5 | Tiedtke 1989, 1993 |
| | Kain-Fritsch | 6 | Kain and Fritsch, 1990; Kain 2004 |
| | MM5 Shallow cumulus (only mixing) (*) | -1 | Grell et al. 1994 |
| **Planetary Boundary Layer** (*ibltyp*) | Modified-Holtslag | 1 | Holtslag et al., 1990 |
| | UW | 2 | Bretherton et al. 2004 |
| **Land Surface** (*code compiling option*) | BATS | / | Dickinson et al. 1993; Giorgi et al. 2003 |
| | CLM4.5 | / | Oleson et al. 2013 |
| **Ocean Fluxes** (*iocnflx*) | BATS | 1 | Dickinson et al. 1993 |
| | Zeng | 2 | Zeng et al. 1998 |
| | COARE | 3 | Fairall et al. 1996a,b |

| Interactive lake (*lakemod*) | 1D diffusion/convection | 1 | Hostetler et al. 1993 |
|---|---|---|---|
| Tropical band (*i_band*) | RegT-Band | 1 | Coppola et al. 2012 |
| Coupled ocean (*iocncpl*) | RegCM-ES | 1 | Sitz et al. 2017 |

**Table 1 Core and sub-grid physics scheme available in RegCM-NH. New schemes**
**available with this release are starred (*).**


**Explicit microphysics schemes**
***Nogherotto-Tompkins Scheme***
A new parameterization for explicit cloud microphysics and precipitation built upon the
European Centre for Medium Weather Forecast's Integrated Forecast System (IFS)
module (Tiedtke [1993], Tompkins [2007]), was introduced in RegCM4 (*ipptls* = 2 in
*µparam*) by Nogherotto et al. [2016]. In the present configuration, the scheme
implicitly solves 5 prognostic equations for water vapor, qv, cloud liquid water, ql, rain, qr,
cloud ice, qi, and snow, qs, but it is also easily extendable to a larger number of variables.
Water vapor, cloud liquid water, rain, cloud ice and snow are all expressed in terms of the
grid-mean mixing ratio.
Cloud liquid and ice water content are independent, allowing the existence of supercooled
liquid water and mixed-phase clouds. Rain and snow precipitate with a fixed terminal fall
speed and can then be advected by the three dimensional winds. A check for the
conservation of enthalpy and of total moisture is ensured at the end of each timestep. The
governing           equation           for           each           variable           is:

$$\frac{\partial q_x}{\partial t} = S_x + \frac{1}{\rho}\frac{\partial}{\partial z}(\rho V_x q_x)$$



The local variation of the mixing ratio qx  of the variable x is given by the sum of
Sx, containing the net sources and sinks of qx  through microphysical processes (i.e.
condensation, evaporation, auto-conversion, melting, etc.), and the sedimentation term,
which is a function of the fall speed Vx . An upstream approach is employed to solve the
equations. The sources and sinks contributors are divided in two groups according to the
duration of the process they describe: processes that are considered to be fast relative to
the model time step are treated implicitly while slow processes are treated explicitly. The
processes taken into account (shown in Figure 2) are the microphysical pathways across
the 5 water variables: condensation, autoconversion, evaporation, cloud water collection
(accretion), and autoconversion for warm clouds, and  freezing, melting, deposition,
sublimation for cold clouds.

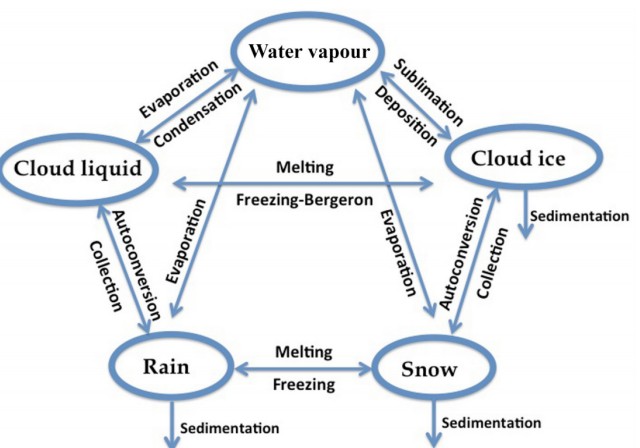


**268   Figure 2: Depiction of the new scheme, showing the five prognostic variables and**
**269   how they are related to each other through microphysical processes**

For each microphysical pathway, phase changes are associated with the release or
absorption of latent heat, which then impacts the temperature budget. The impact is
calculated using the conservation of liquid water temperature TL defined as:

$$T_L = T - \frac{L_v}{C_p}(q_l + q_r) - \frac{L_s}{C_p}(q_i + q_s).$$


Given that dTL =0, the rate of change of the temperature is given by the following
equation:

$$\frac{\partial T}{\partial t} = \sum_{x=1}^{m} \frac{L(x)}{C_p}\left(\frac{dq_x}{dt} - D_{q_x} - \frac{1}{\rho}\frac{\partial}{\partial z}(\rho V_x q_x)\right)$$


where $L(x)$ is the latent heat of fusion or evaporation, depending on the process
considered, $Dqx$ is the convective detrainment and the third term in brackets is the
sedimentation term.
At the end of each time step a check is carried out of the conservation of total water and
moist static energy: $h = C_P T + gz + Lq_x.$
The scheme is tunable through parameters in the *µparam* section of the namelist
(RegCM-4.7.1/Doc/README.namelist; Elguindi et al. 2017).

***WSM5 Scheme***
RegCM4-NH also employs the Single-Moment 5-class microphysics scheme of the WRF
model (Skamarock et al., 2008). This scheme (ipptls = 3 in µparam) follows Hong
et al. (2004) and, similarly to Nogherotto et al. (2016), includes vapor, rain, snow, cloud
ice, and cloud water hydrometeors. The scheme separately treats ice and water
saturation processes, assuming water hydrometeors for temperatures above freezing,
and cloud ice and snow below the freezing level (Dudhia, 1989, Hong et al., 1998). It
accounts for supercooled water and a gradual melting of snow below the melting layer
(Hong et al., 2004, and Hong and Lim, 2006). Therefore, the WSM5 and Nogherotto-
Tompkins schemes have similar structures (Figure 2), but also important differences.
Differently from the Nogherotto-Tompkins scheme, the WSM5 (as well as the other WSM
schemes in WRF) prescribes an inverse exponential continuous distribution of particle
size (ex. Marshall and Palmer (1948) for rain, Gunn and Marshall (1958) for snow). It also
includes the size distribution of ice particles and, as a major novelty, the definition of the
number of ice crystals based on ice mass content rather than temperature. Both the
Nogherotto-Tompkins and WSM5 schemes include autoconversion, i.e. sub-time step
processes of conversion of cloud water to rain and cloud ice to snow. For rain, Hong et
al. (2004) use a Kessler (1969) type algorithm in WSM5, but with a stronger physical basis
following Tripoli and Cotton (1980). The Nogherotto-Tompkins scheme also includes the
original Kessler (1969) formula as an option, but it makes available other three
exponential approaches following Sundqvist et al. (1989), Beheng (1994), and
Khairoutdinov and Kogan (2000). For ice autoconversion the Nogherotto-Tompkins
scheme uses an exponential approach (Sundqvist, 1989) with a specific coefficient for ice
particles (following Lin et al., 1983) depending on temperature, while the WSM5 uses a
critical value of ice mixing ratio (depending on air density) and a maximum allowed ice
crystal mass (following Rutledge and Hobbs, 1983) that suppresses the process at low
temperatures because of the effect of air density. Finally, the WSM5 has no dependency
on cloud cover for condensation processes while the Nogherotto-Tompkins scheme uses
cloud cover to regulate the condensation rate in the formation of stratiform clouds.
*Illustrative case studies*

Three case studies (Table 2) of Heavy Precipitation Events (HPE) have been identified in
order to test and illustrate the behavior of the non-hydrostatic core of the RegCM4-NH,
with focus on the explicit simulation of convection over different regions of the world. In
two of the test cases, California and Lake Victoria, data from the ERA-Interim reanalysis
(Dee et al. 2011) are used to provide initial and lateral meteorological boundary conditions
(every 6 hours) for an intermediate resolution run (grid spacing of 12 km, with use of
convection parameterizations), which then provides driving boundary conditions for the
convection-permitting experiments (Figure 3). In the Texas case study, however, we
nested the model  directly in the ERA-Interim reanalysis given that  such configuration
was able to  accurately reproduce the HPE intensity. In this case the model uses a large
LBC relaxation zone which allows the description of realistic fine-scale features driving
this weather event (although not fully consistent with the Matte et al. (2017) criteria). All
simulations start 24-48 hours before the HPE (Table 2). The analysis focuses on the total
accumulated precipitation over the entire model domain at 3 km resolution (Figure 2) for
the periods defined in Table 2. In the cases of California and Texas  the evaluation also
includes the time series of 6 hourly accumulated precipitation averaged on the region of
maximum precipitation (black  rectangles  in Figures 5a and 7a) because high temporal
resolution observations (NCEP/CPC) are also available (Table 3). The discussion of the
case studies is presented in the next sections; the configuration files (namelists) with full
settings for the three test cases are available at https://zenodo.org/record/5106399.

A key issue concerning the use of CP-RCMs is the availability of very high resolution,
high quality observed datasets for the assessment and evaluation of the models, which
is lacking for most of the world regions. Precipitation measurements come from
essentially three distinct sources: in-situ rain-gauges, ground radar and satellite. In the
present study we use 7 observational datasets depending on the case study and the area
covered, as described in Table 3. We have used: Precipitation Estimation from Remotely
Sensed Information using Artificial Neural Networks - Climate Data Record (PERSIAN-
CDR), Climate Hazards Group InfraRed Precipitation with Station data (CHIRPS),  the
Climate Prediction Center morphing method (CMORPH), Tropical Rainfall Measuring
Mission (TRMM), NCEP/CPC-Four Kilometer Precipitation Set Gauge and Radar
(NCEP/CPC), CPC-Unified gauge-based daily precipitation estimates (CPC) and
Parameter-elevation Regressions on Independent Slopes Model (PRISM) (Table 3).
NCEP/CPC is a precipitation analysis which merges a rain gauge dataset with radar
estimates. CMORPH and PERSIAN-CDR are based on satellite measurements, CHIRPS
incorporates satellite imagery with in-situ station data. CPC is a gauge-based analysis of
daily precipitation. The PRISM dataset gathers climate observations from a wide range
of monitoring networks, applying sophisticated quality control measures and developing
spatial climate datasets which incorporate a variety of modeling techniques at multiple
spatial and temporal resolutions.

| Case | ACRONYM | Region of The event | Domains size lon x lat x vertical levels | Simulation Time Window (UTC) |
|------|---------|---------------------|------------------------------------------|------------------------------|
| 1 | CAL | California | 480 x 440 x 41 | 15 Feb 2004 00:00 19 Feb 2004 00:00 |
| 2 | TEX | Texas | 480 x 440 x 41 | 9 June 2010 00:00 12 June 2010 00:00 |
| 3 | LKV | Lake Victoria | 550 x 530 x 41 | 25 Nov 1999 00:00 1 Dec 1999 00:00 |

**Table 2: List of acronyms and description of the test cases with corresponding**
**3km domain sizes and simulation period.**

| Dataset name | Region | Spatial Resolution | Temporal Resolution | Data Source | Reference |
|--------------|--------|--------------------|--------------------|-------------|-----------|
| TRMM | World | 0.5° | Daily | Satellite | Huffman et al. (2007) |

| CHIRPS | World | 0.05° | Daily | Station data+Satellite | Funk et al. (2015) |
|---|---|---|---|---|---|
| CMORPH | World | 0.25° | Daily | Satellite | Joyce et al. (2004) |
| NCEP/CPC | USA | 0.04° | Hourly | *Gauge and Radar* | https://doi.org/10.5065/D69Z93M3. Accessed: 27/06/2018 |
| CPC | World | 0.5° | Daily | Station data | Chen and Xie (2008) |
| PRISM | USA | 0.04° | Daily | Station data | PRISM Climate Group. 2016. |
| PERSIAN-CDR | World | 0.25° | Daily | Satellite | Ashouri et al. (2015) |

**Table 3: List of observed precipitation datasets used for comparison.**

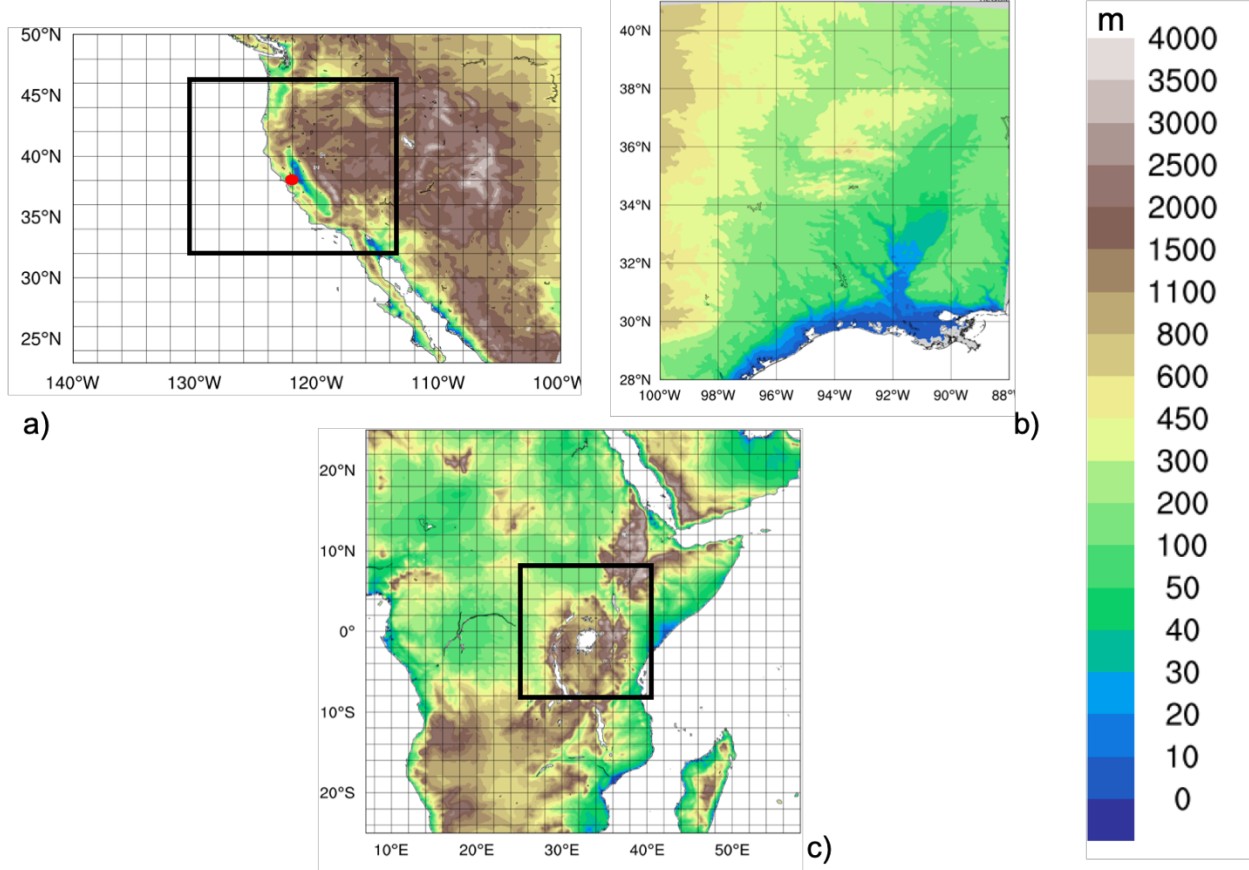


**Figure 3: Domains tested , a) California (CAL) , b) Texas (TEX), c) Lake Victoria (LKV) . For CAL (a) and LKV (b) the black square shows the 3 km simulation domains nested in the 12 km domain in figure. For TEX case (b) the 3 km domain simulation is fed directly with the ERA-Interim reanalysis fields.**

370

371

## California

The first case, referred to as CAL in Table 2, is a HPE which occurred on February 16-18 2004, producing flooding conditions for the Russian River, a southward-flowing river in the Sonoma and Mendocino counties of northern California (red-dot in Figure 3a). The event is documented in detail by Ralph et al. (2006), who focused their attention on the impact of narrow filament-shaped structures of strong horizontal water vapor transport

over the eastern Pacific Ocean and the western U.S. coast, called Atmospheric Rivers
(ARs). ARs are typically associated with a low-level jet stream ahead of the cold front of
extratropical cyclones (Zhu and Newell 1998; Dacre et al. 2015; Ralph et al. 2018), and
can induce heavy precipitation where they make landfall and are forced to rise over
mountain chains (Gimeno et al. 2014). The CAL event consists of a slow propagating
surface front arching southeastward towards Oregon and then southwestward offshore
of California (Figure 4a,c). Rain began over the coastal mountains of the Russian River
watershed at 0700 UTC of February 16, as a warm front descended southward, and also
coincided with the development of orographically favoured low-level upslope flow (Ralph
et al., 2006).

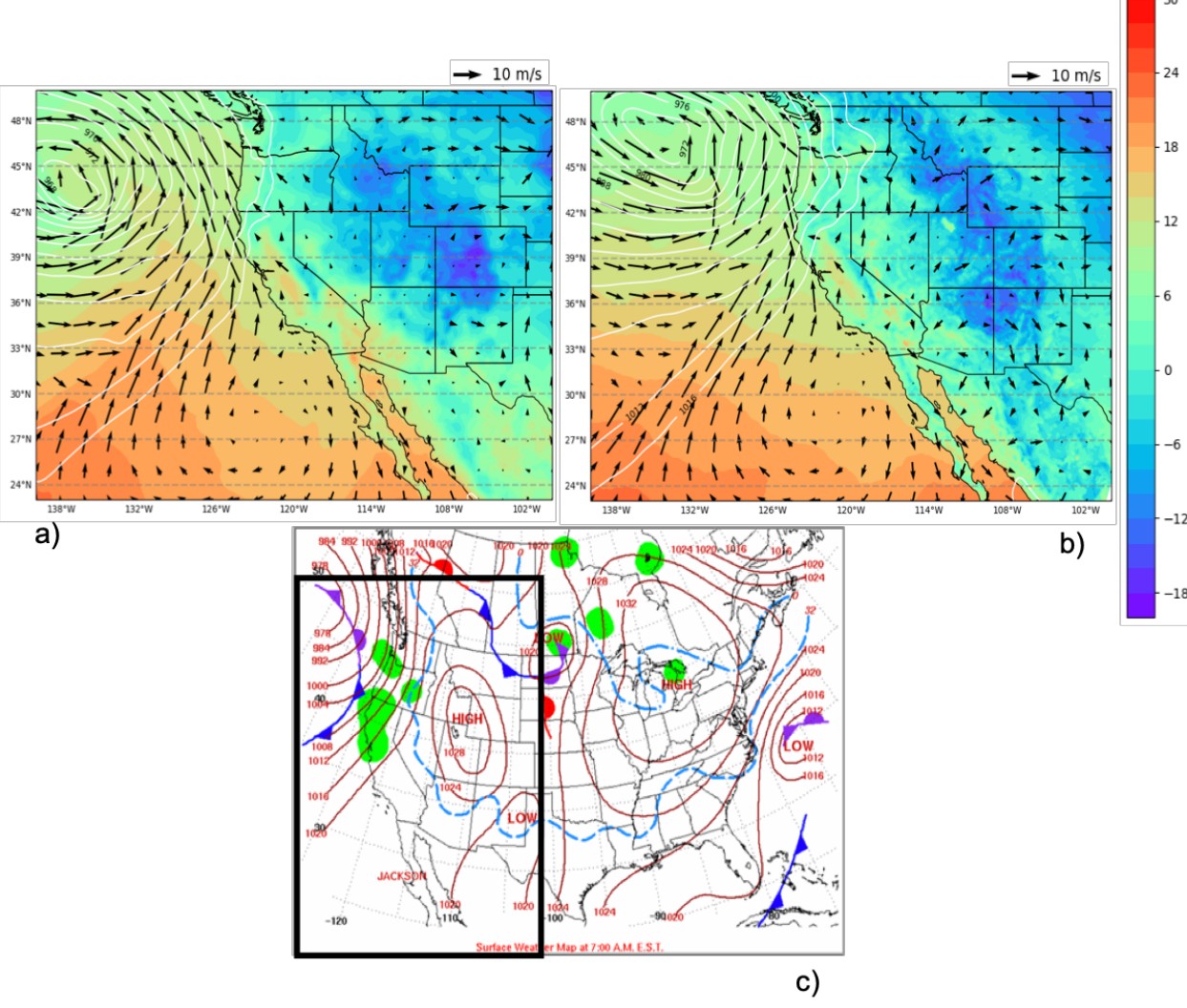


**Figure 4: a,b) mean sea level pressure (mslp, hPa, white contour lines), surface temperature (color shading, °C) and 100-m wind direction (black arrows, m/s) at 0700 UTC, February 16, 2004 of ERA5 reanalysis and RegCM 12km respectively. c) NCEP-NOA Surface Analysis of pressure and fronts. The black box in (c) bounded the area represented in (a) and (b)**

The intermediate resolution (12 km) domain (Figure 3a) covers a wide area encompassing California and a large portion of the coastal Pacific Ocean, with 23 vertical levels and a parameterization for deep convection based on the Kain–Fritsch scheme (Kain, 2004). The ERA-Interim driven simulation is initialized at 0000 UTC, February 15 2004 (Table 2) and lasts until 0000 UTC February 19 2004. This simulation is used as a boundary conditions for a RegCM4-NH run over a smaller area centered over northern California (Fig. 3a) at 3 km horizontal resolution, with 41 vertical levels and boundary conditions updated every 6 hours. In RegCM4-NH only the shallow convection code of the Tiedtke scheme (Tiedtke, 1996) is activated. Simulated precipitation is compared with the CHIRPS, CMORPH, TRMM, PRISM, NCEP/CPC observations (Table 3).

As shown in Figure 4 the February 16 synoptic conditions for mean sea level pressure (mslp), surface temperature and wind direction of this case study, are well reproduced by RegCM4 at 12 km (Fig. 4b) when compared to ERA5 reanalysis (Fig. 4a). The surface analysis of pressure and fronts derived from the operational weather maps prepared at the National Centers for Environmental Prediction, Hydrometeorological Prediction Center, National Weather Service (https://www.wpc.ncep.noaa.gov/dailywxmap/index_20040216.html) is also reported in Figure 4c.

The available observed precipitation datasets show similar patterns for the total accumulated precipitation (Figure 5), in particular CHIRPS (Figure 5a), PRISM (Figure 5d) and NCEP (Figure 5e) exhibit similar spatial details and magnitudes of extremes. CHIRPS shows a maximum around 42°N which is not found in the other datasets. CMORPH (Figure 5b) and TRMM (Figure 5c) show lower precipitation maxima and lesser spatial details due to their lower resolution, indicating that the performance of satellite-

based products may be insufficient as a stand alone product to validate the model for this
case.
The largest observed maxima are placed on the terrain peaks, with extreme rainfall
greater than 250 mm in 60 hours over the coastal mountains and between 100 – 175 mm
elsewhere (Fig. 5). The black box in Fig 5a shows the area of the Russian River
watershed where the largest rainfall rates were detected (269 mm and 124 mm in 60-h
accumulated rainfall between 0000 UTC February 16 and 1200 UTC February 18, 2004,
respectively) (Ralph et al., 2006).
The convection-permitting simulation captures the basic features of the observed
precipitation, both in terms of spatial distribution (Fig. 5f) and of temporal evolution of
rainfall (Fig. 6a). However, it shows higher precipitation rates than observed over the sea
and over the mountain chains, with lower intensities than observed in the south-east part
of the mountain chain (Fig. 5). The 12-km simulation instead severely underestimates the
magnitude of the event (Fig. 5g).
Figure 6a shows the 6-hourly accumulated precipitation averaged over the black box in
Figure 5a. The 3 km and 12 km simulations capture the onset of the event, but the peak
intensity is strongly underestimated by the 12 km run, while it is well simulated by the 3
km run, although the secondary maximum is overestimated. These results demonstrate
that only the high resolution convection-permitting model is able to captures this extreme
event, and that parameterized convection has severe limits in this regard (Done et al.
2004; Lean et al. 2008; Weisman et al. 2008; Weusthoff et al. 2010; Schwartz 2014; Clark
et al. 2016).

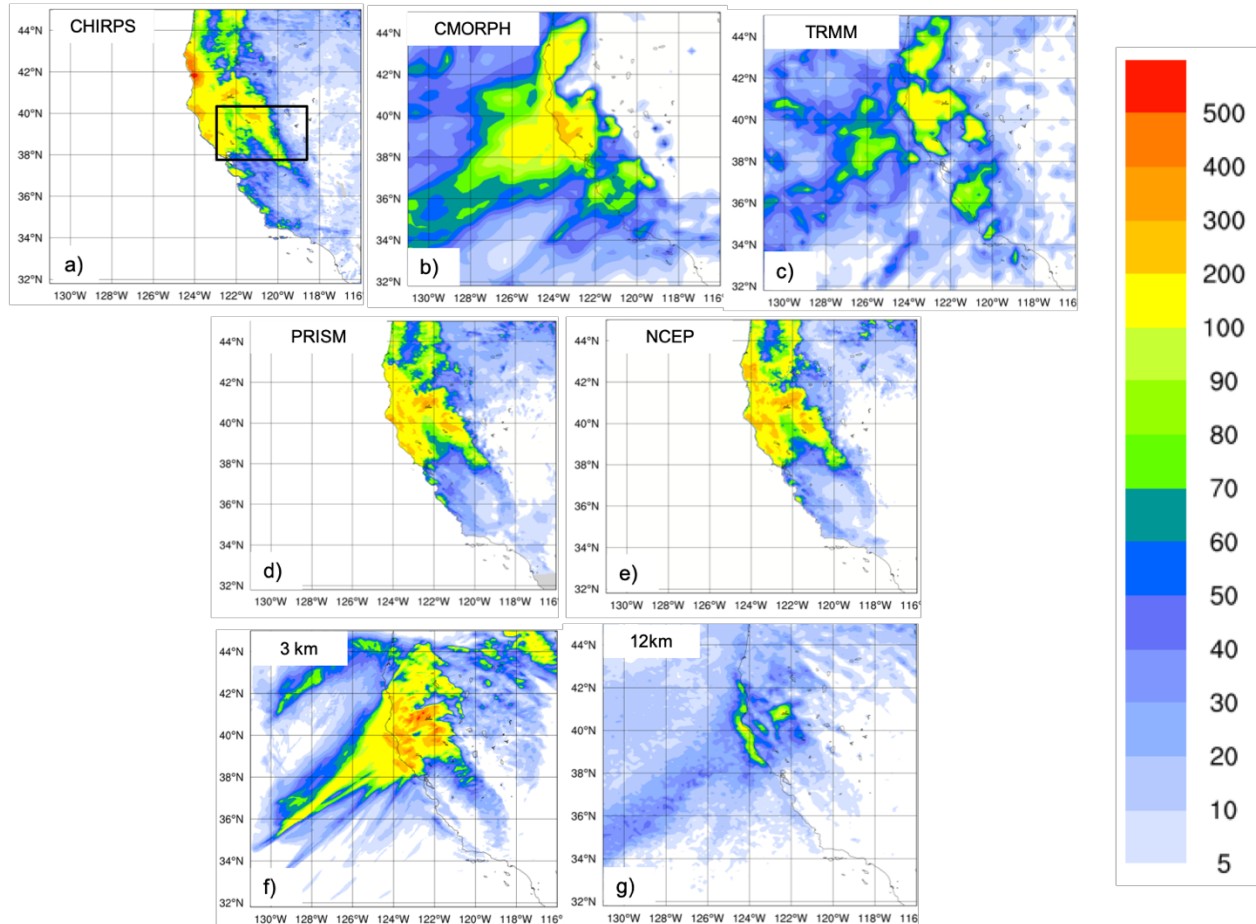


**Figure 5 : Total accumulated precipitation (mm) during the California case: CHIRPS (a), CMORPH (b), TRMM (c) observations, PRISM (d) and NCEP Reanalysis (e) and convection-permitting simulation with RegCM4-NH at 3km (f) and RegCM4 at 12km (g). The black box denotes the area where the spatial average of 6-hourly accumulated precipitation is calculated for Figure 6a.**






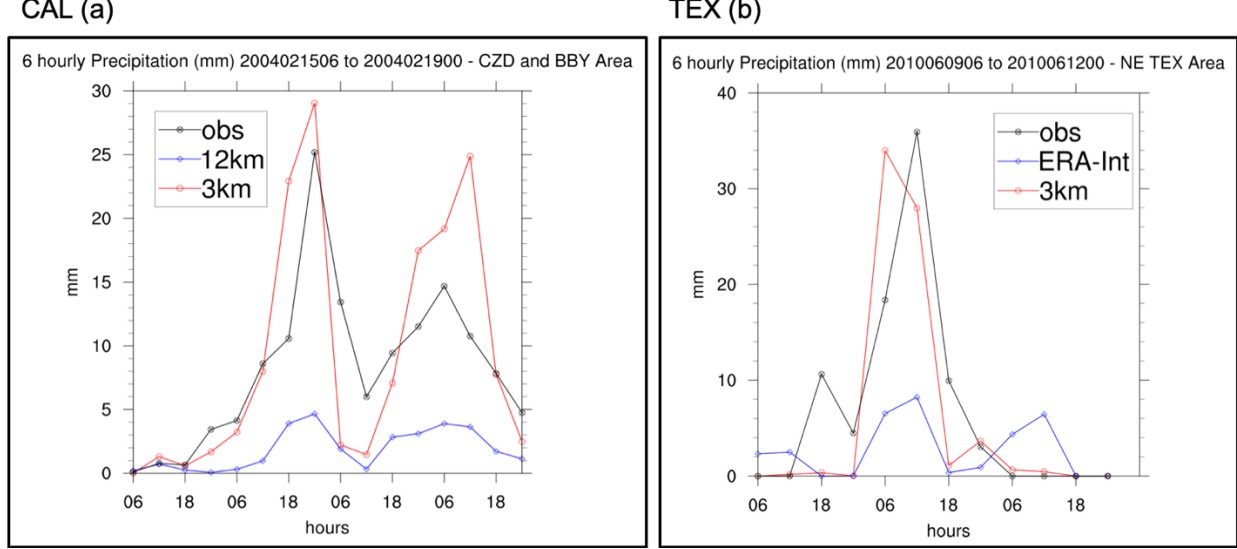


**Figure 6: Time series of the 6 hourly accumulated precipitation (in mm on the y-axis) during the CAL event (a) and during the TEX event (b). The blue lines show RegCM4 12 Km and ERA interim 6 hourly accumulated precipitation averaged over the areas indicated by the black squares in Figures 5 and 7 while the red line shows the 6 hourly accumulated precipitation simulated by RegCM4-NH. The observations are shown with a black line.**

## Texas

Case 2, hereafter referred to as TEX (Table 2), is a convective precipitation episode exhibiting characteristics of the "Maya Express" flood events, linking tropical moisture plumes from the Caribbean and Gulf of Mexico to midlatitude flooding over the central United States (Higgins 2011). During the TEX event, an upper-level cutoff low over northeastern Texas, embedded within a synoptic-scale ridge, moved slowly northeastward. Strong low-level flow and moisture transport from the western Gulf of Mexico progressed northward across eastern Texas. The event was characterized by low-level moisture convergence, weak upper-level flow, weak vertical wind shear, and relatively cold air (center of cutoff low), which favored the slow-moving convective storms and nearly stationary thunderstorm outflow boundaries. The main flooding event in eastern Texas occurred on June 10, 2010, with a daily maximum rainfall of 216.4 mm for the region in the black box of Figure 7a (Higgins 2011).

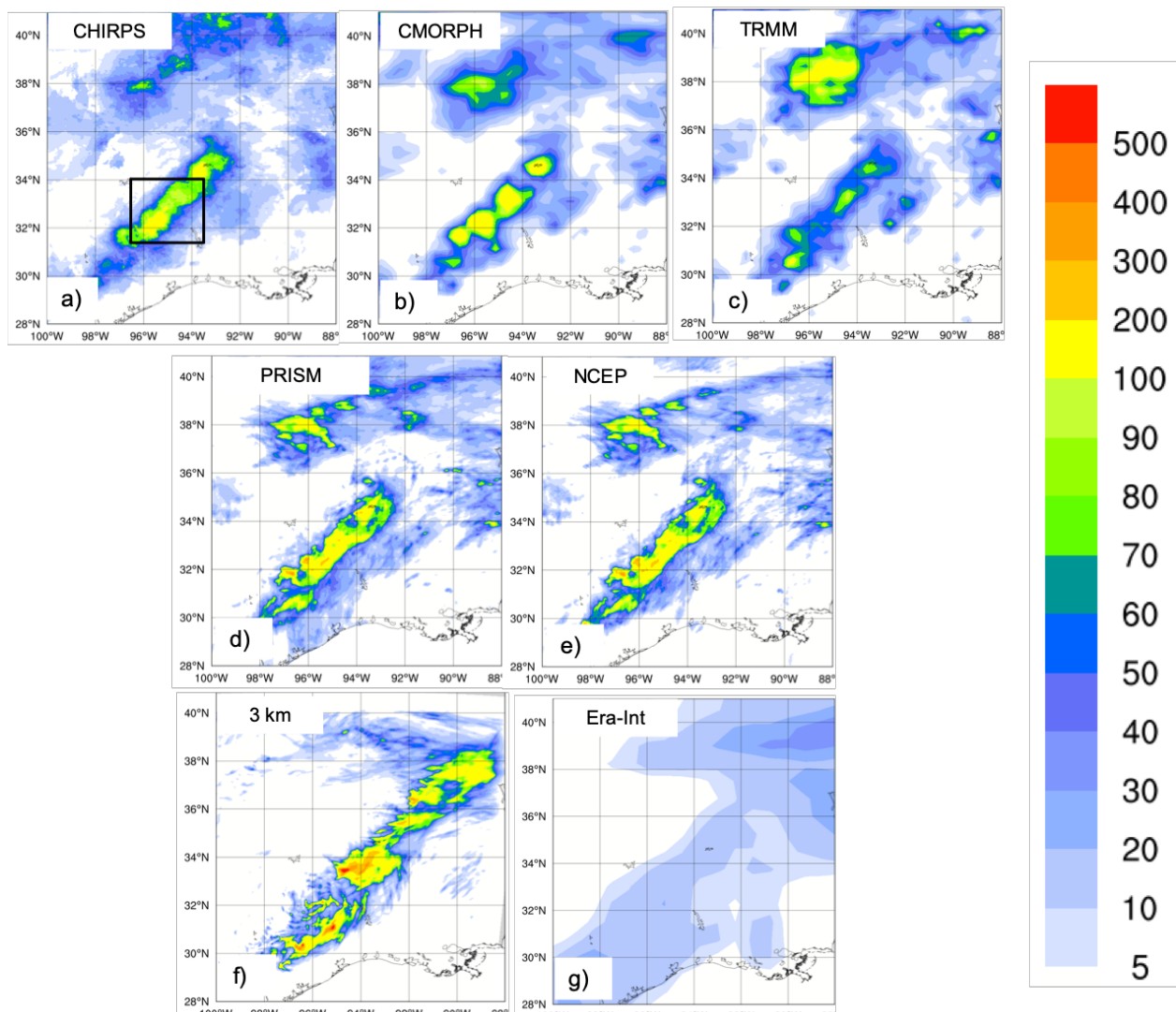

471

**Figure 7: Total accumulated precipitation (mm) during the Texas case: CHIRPS (a), CMORPH (b), TRMM (c), PRISM (d), NCEP Reanalysis (e) and convection-permitting simulation with RegCM4-NH at 3 km grid spacing (f) and ERA-Interim (g). The black box (a) shows the area where the spatial average of 6-hourly accumulated precipitation was calculated for Figure 6b**

As for the California case, the observed precipitation datasets show coherent patterns for the total accumulated precipitation (Figure 7), with the highest values related to the mesoscale convective system in eastern Texas (~ 200 mm), and another smaller area of high precipitation more to the north, approximately over Oklahoma. PRISM (Figure 7d)and NCEP (Figure 7e) capture similar spatial details and magnitudes of extremes,

CHIRPS (Figure 7a) has lower precipitation extremes in the north compared to the other
datasets, while CMORPH (Figure 7b) and TRMM (Figure 7c) show the lowest
precipitation extremes and reduced spatial details as already noted for the California
case.
Figure 7f and Figure 7g present precipitation as produced by the RegCM4-NH and the
ERA-Interim reanalysis (driving data) respectively. ERA-Interim reproduces some of the
observed features of precipitation, but with a substantial underestimation over the areas
of maximum precipitation because of its coarse resolution. By comparison, the RegCM4-
NH simulation (Fig. 7f) shows an improvement in both pattern and intensity of
precipitation, and is substantially closer to observations over eastern Texas. However,
the precipitation area is slightly overestimated and the model is not capable of
reproducing the small region of maximum precipitation in the north.

The time series of precipitation over eastern Texas from June 9 to 12, 2010 for
observations (black line), ERA-Interim (blue line) and RegCM4-NH (red line) are reported
in figure 6b. Precipitation increases over this region from 0000 UTC until it reaches the
observed maximum at 1200 UTC, on June 10 (~35 mm), gradually decreasing afterwards
until 0600 UTC, on June 11. The RegCM4-NH simulation shows a more realistic temporal
evolution than the ERA-Interim, which exhibits an overall underestimation of precipitation.
The non-hydrostatic model produces precipitation values closer to the observations,
however the simulated maximum is reached 6 hours earlier than observed.


**Lake Victoria**
Case 3 focuses on Lake Victoria (LKV), with the purpose of testing RegCM4-NH on a
complex and challenging region in terms of convective rainfall. It is estimated that each
year 3,000-5,000 fishermen perish on the lake due to nightly storms (Red Cross, 2014).
In the Lake Victoria basin, the diurnal cycle of convection is strongly influenced by
lake/land breezes driven by the thermal gradient between the lake surface and the
surrounding land. As the land warms during the course of the day, a lake breeze is
generated which flows from the relatively cooler water towards the warmer land surface.
The circulation is effectively reversed at night, when the land surface becomes cooler
than the lake surface, leading to convergence over the lake and associated thermal
instability.
In the LKV region, prevailing winds are generally easterly most of the year with some
variability due to the movement of the ITCZ. The local diurnal circulation created by the
presence of the lake creates two diurnal rainfall maxima. During daylight hours, when the
lake breeze begins to advance inland, convergence is maximized on the eastern coast of
the lake as the lake breeze interacts with the prevailing easterlies. Studies have also
noted the importance of downslope katabatic winds along the mountains to the east of
the lake in facilitating convergence along the eastern coastal regions (Anyah et al. 2006).
This creates a maximum in rainfall and convection on the eastern coast of LKV.
Conversely, during nighttime hours, when the local lake circulation switches to flow from
the land towards the lake, the prevailing easterlies create locally strong easterly flow
across the lake and an associated maximum in convergence and rainfall on the western
side of LKV.
The LKV simulation starts on November 25, 1999 and extends to the beginning of
December 1999 (Table 2), covering a 5-day period which falls within the short-rain season
of East Africa. The choice of 1999, an ENSO neutral year, was made in order to focus the
analysis on local effects, such as the diurnal convection cycle in response to the lake/land
breeze, with no influence of anomalous large scale conditions. A 1-dimensional lake
model (Hostetler et al. 1993; Bennington et al. 2014) interactively coupled to RegCM4-
NH was utilized to calculate the lake surface temperature (LST), since lake-atmosphere
coupling has been shown to be important for LKV (Sun et al. 2015; Song et al. 2004).
This coupled lake model has been already used for other lakes, including Lake Malawi in
southern Africa (Diallo et al. 2018). As with the other experiments, the boundary
conditions are provided by a corresponding 12 km RegCM4 simulation employing the
convection scheme of Tiedtke (1996).
At the beginning of the simulation, the LST over the lake is uniformly set to 26°C, and is
then allowed to evolve according to the lake-atmosphere coupling. This initial LST value
is based on previous studies. For example, Talling (1969) finds Lake Victoria surface
temperatures ranging from 24.5-26°C during the course of the year. Several studies have
used RCMs to investigate the Lake Victoria climate (Anyah et al., 2006; Anyah and
Semazzi 2009, Sun et al. 2015), and found a significant relationship between lake
temperature and rainfall depending on season. The value of 26°C is typical of the winter
season and was chosen based on preliminary sensitivity tests using different values of
initial temperature ranging from 24°C to 26°C.
The synoptic feature favorable for the production of precipitation over the LKV in this
period corresponds to a large area of southeasterly flow from the Indian Ocean (Fig. 8a),
which brings low-level warm moist air into the LKV region facilitating the production of
convective instability and precipitation. This synoptic situation, with a low-level south-
easterly jet off the Indian Ocean, is a common feature associated with high precipitation
in the area (Anyah et al. 2006), and can be seen in ERA5 data (Figure 8a). Although
some bias in terms of magnitude, this is reasonably well reproduce by the 12 km
simulation (Figure 8b).

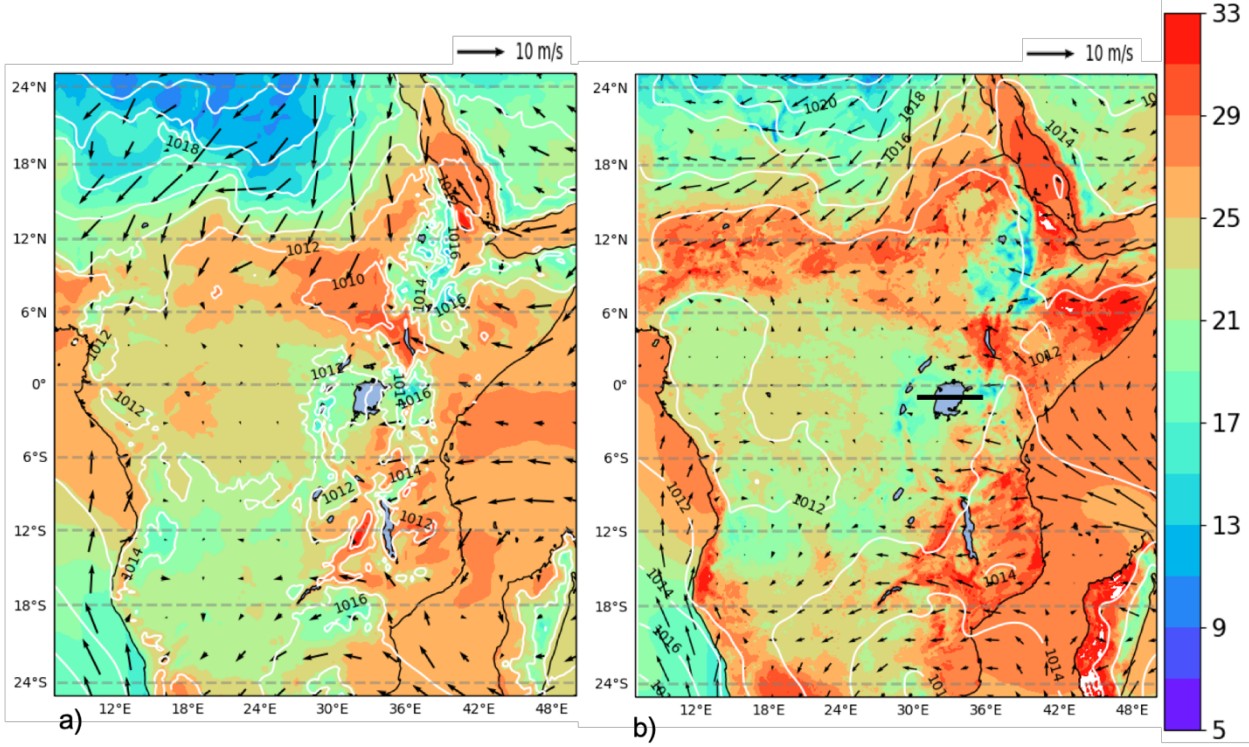


**Figure 8: Mean sea level pressure (mslp) (hPa) (white contour lines), surface temperature**
**(color shading) (°C) and 100-m wind (black arrows) averaged over the period 25 November**

**0000 UTC - 1 December 0000 UTC, by ERA5 reanalysis (a) and RegCM 12km (b). The black line (b) shows the cross-section position represented in Fig. 9**

The LKV region dynamics are quite distinct between nighttime and daytime and the rainfall in and around the lake has a pronounced diurnal cycle. To understand this strong diurnal cycle, Figure 9 shows a cross-section through the lake (32E to 34E, black line in right panel of Fig. 8b) along 1°S latitude at a period during strong nighttime (Fig. 9b,d; 0600Z November 30) and daytime convection (Figure 9a,c; 12Z November 29). Wind vectors in Figure 9 show the zonal-wind anomaly across 0°-2°S to highlight the circulations associated with LKV. During the day, surface heating around the lake leads to a temperature difference between the land and lake sufficient to generate a lake breeze, which causes divergence over the lake, while over the highlands to the east the environment is more conducive to convection where convergence is focused (9a,c). Conversely, during the night, a land breeze circulation is generated, which induces convergence and convection over the lake (Figure 9b,d). In Figure 10, the evolution of the nighttime land breeze is illustrated with cooler temperature anomalies propagating westward onto the lake during the night.

Comparing the 3 km simulation to the 12 km forcing run, we find that the localized circulations created by local forcings (i.e. convection) are much stronger in the convection permitting resolution experiment. We also find stronger and more localized areas of convective updrafts compared to the 12 km simulation (Figure 9c,d; omega is shown instead of vertical velocity here because of the difference in dynamical core).  As an example during the nighttime event (Figure 9b,d)  there is a broad area of upward motion over the lake and the associated broad convergence in the 12km simulation, while in the convection permitting 3km simulation, convection is much more local and concentrated over the western part of the lake. Indeed, nighttime rainfall tends to be concentrated over the western part of the lake ( Sun et al. 2015; Figure 11a-d). Stronger convection simulated in the 3 km experiment could also be tied to  stronger temperature anomalies shown over the lake and land and between day and night relative to the 12km simulation (Figure 10). The 3km simulation also shows a more pronounced land breeze propagation at night compared to the 12km simulation.

This demonstrates that the 3km simulation is better equipped to simulate the localized
circulations associated with this complex land-lake system.

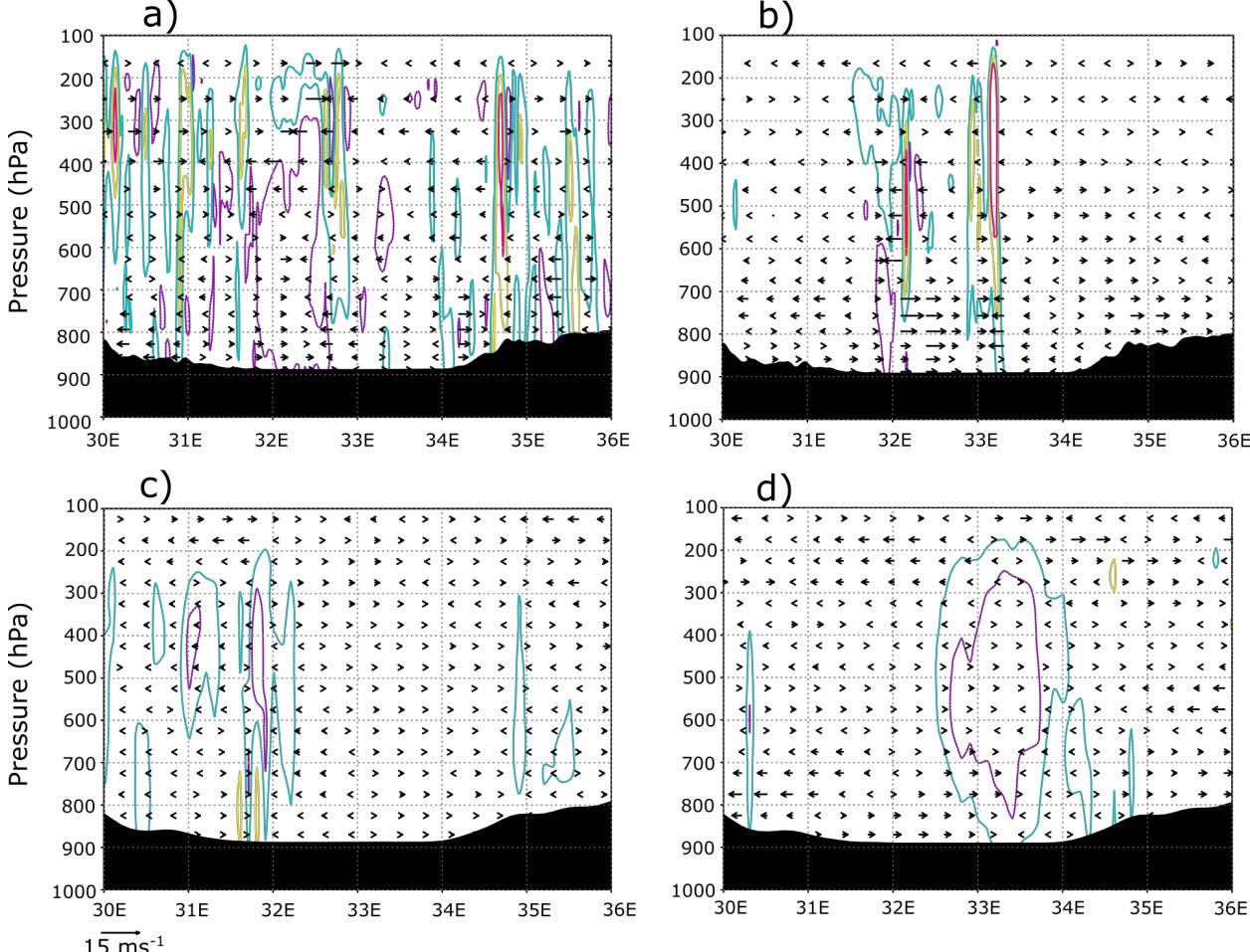



**Figure 9. Cross-section through 1°S (black line in Fig. 8b) of the  zonal-wind anomaly (0°-**
**2°S) vectors and the mean contoured vertical velocity (m/s) over 0°-2°S at a) 12Z 29**
**November and b) 6Z 30 November from the 3km simulation. Purple dashed contours**
**indicate -0.1 m/s, light blue contours indicate 0.1 m/s, yellow contours indicate 0.3 m/s,**
**and red contours indicate 0.5 m/s. Lake Victoria encompasses about 32°E to 34°E. The**
**bottom 2 panels show the same as in a) and b) but from the 12km simulation at c) 12Z 29**
**November and d) 6Z 30 November. Purple dashed contours indicate -0.01 hPa/s, light blue**
**dashed contours indicate -0.005 hPa/s, and yellow dashed contours indicate 0.005 hPa/s.**

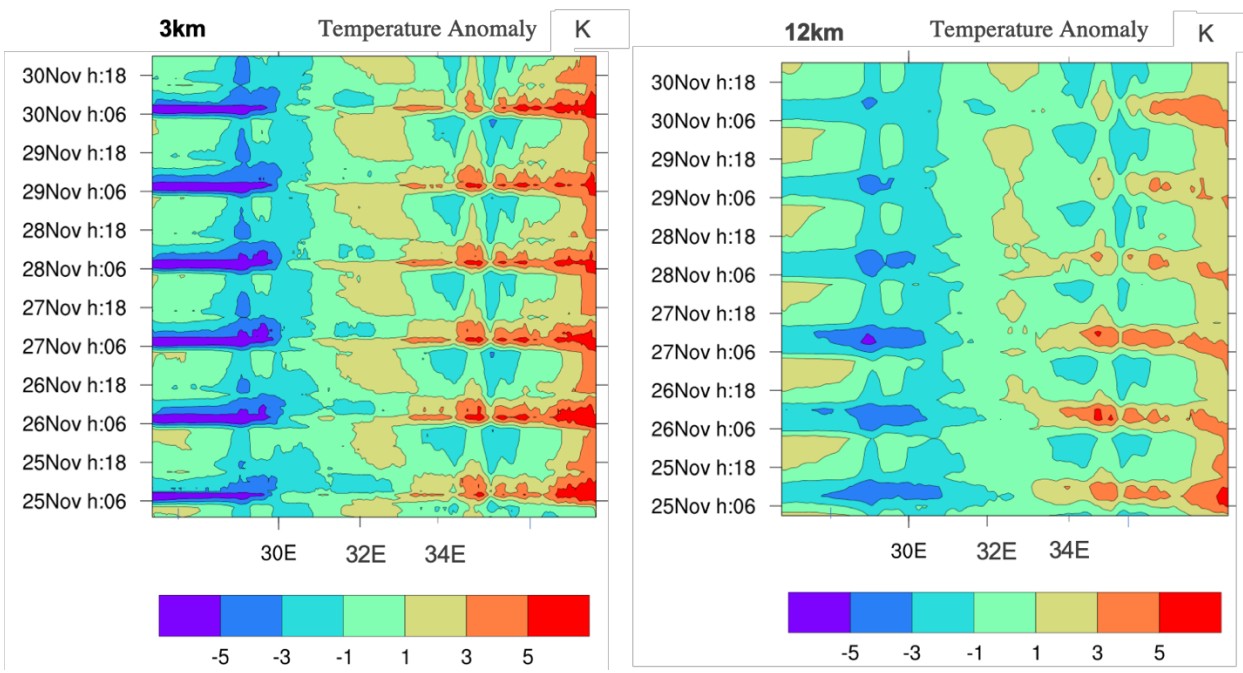

**Figure 10** : **Longitude-time (hourly) Hovmöller diagram of LKV domain surface temperature anomaly (shading, in K). Panels correspond to the 3km simulation (left) and 12km simulation (right).  The lake Victoria is between 32°E and 34°E longitude**

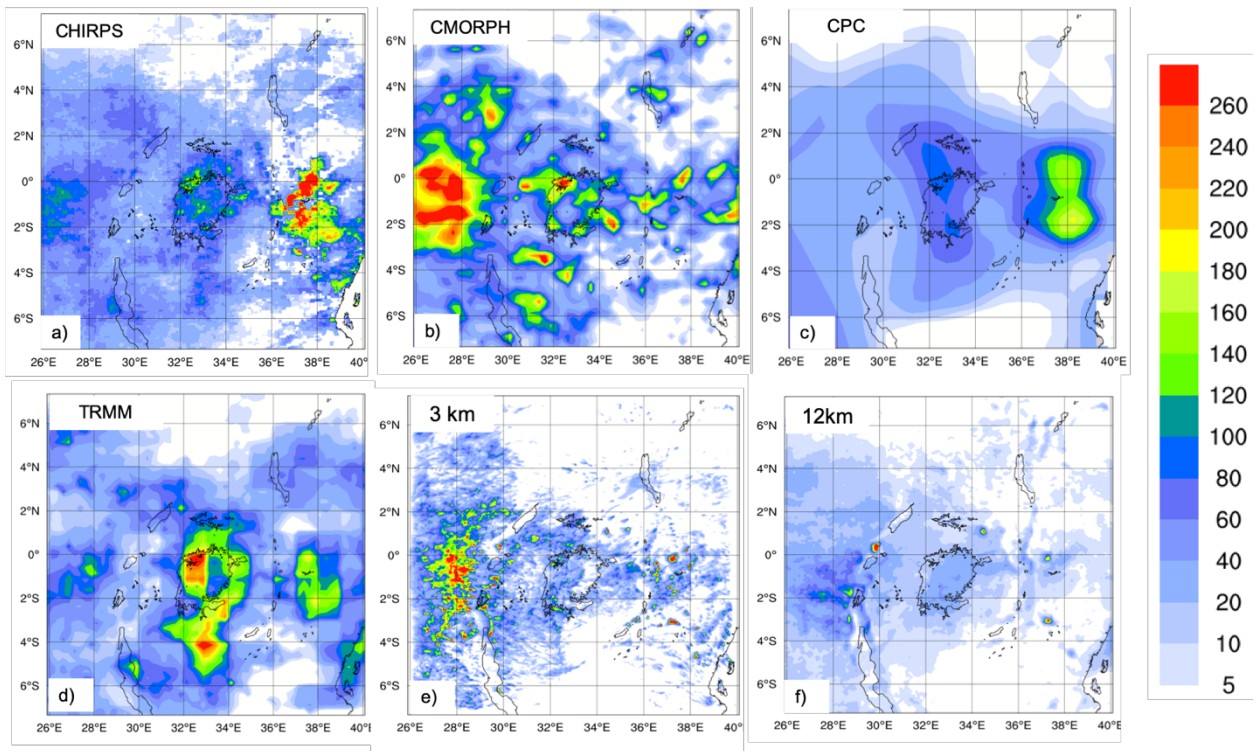

608

**Figure 11: Total event accumulated precipitation (mm) during the LKV case (November 25, 1999-December 1, 1999) measured by CHIRPS (a), CMORPH (b), CPC (d) TRMM (e) and calculated by RegCM4 at 3 km (e) and 12 km (f).**

612

Figure 11 reports the total accumulated precipitation observed and simulated for the LKV case. TRMM (Figure 11d) and CPC (Figure 11c) show a similar pattern, with two-rainfall maxima of different intensities over the southeastern and northwestern lake areas. CMORPH (Figure 11b) shows a western rainfall maximum similar to TRMM and one large rainfall area almost entirely centered over the highlands to the west of the lake. Conversely in CHIRPS (Figure 11a) a maximum is found to the east of the lake while several localized maxima occur over the lake. The differences among the observed datasets highlight  the issue of observational uncertainty and the need to take into consideration shortcomings associated with the types of observational datasets considered. Different datasets can have significantly different climatologies, especially in areas of low data availability. For example, Prein and Gobiet (2017) analyzed two gauge-based European-wide datasets, and seven global low-resolution datasets, and found

large differences across the observation products, often of similar magnitude as the
difference among model simulations. In this case and for this area the observation
uncertainty plays a big role especially at high resolution, and highlights the need for an
adequate observational network for model validation. However, despite the large
uncertainty among the different observed datasets (Figure 11 a-d), we find a significant
underestimation of the precipitation by the 12 km run over the lake independently of the
dataset used as a reference (Figure11f). In contrast, the 3 km simulation (Figure 11e)
shows substantially greater detail, with rainfall patterns more in agreement with the
CMORPH data. In particular, the 3 km simulation reproduces well the local rainfall
maxima on the western side of the lake, although these appear more localized and with
a multi-cell structure compared to CMORPH and TRMM. Additionally, the 12 km
simulation underestimates the observed heavy rainfall totals in the highlands to the west
of the lake region especially when compared to CMORPH, which are instead reproduced
by the 3 km simulation.
This last test case demonstrates the ability of RegCM4-NH in simulating realistic
convective activity over a such morphologically complex region, which is a significant
improvement compared to the hydrostatic-coarse resolution model configuration.

**Conclusions and future outlook**

In this paper we have described the development of RegCM4-NH, a non hydrostatic
version of the regional model system RegCM4, which was completed in response to the
need of moving to simulations at convection-permitting resolutions of a few kilometers.
The non-hydrostatic dynamical core of MM5 has been incorporated into the RegCM4
system previously based on the MM5 hydrostatic core. Some modifications to the MM5
dynamical core were also implemented to increase the model stability for long term runs.
RegCM4-NH also includes two explicit cloud microphysics schemes needed to explicitly
describe convection and cloud processes in the absence of the use of cumulus
convection schemes. Finally, we presented a few case studies of explosive convection to
illustrate how the model provides realistic results in different settings and general
improvements compared to the coarser resolution hydrostatic version of RegCM4 for
such types of events.

As already mentioned, RegCM4-NH is currently being used for different projects, and
within these contests, is being run at grid spacings of a few kilometers for continuous
decadal simulations, driven by reanalyses of observations or GCM boundary conditions
(with the use of an intermediate resolution domains) over different regions, such as the
Alps, the Eastern Mediterranean, Central-Eastern Europe and the Caribbeans. These
projects, involving multi-model inter-comparisons, indicate that the performance of
RegCM4-NH is generally in line with that of other convection-permitting models, and
exhibits similar improvements compared to coarser resolution models, such as a better
simulation of the precipitation diurnal cycle and of extremes at hourly to daily time scales.
The results obtained within the multi-model context confirm previous results from single-
model studies  (Kendon et al. 2012, 2017, Ban et al. 2014, 2015; Prein et al. 2015, 2017),
but also strengthen the robustness of the findings through reduced uncertainty compared
to coarse resolution counterpart (Ban et al., 2021, Pichelli et al., 2021). The convection-
permitting scale can thus open the perspective of more robust projections of future
changes of precipitation, especially over sub-daily time scales.

One of the problems of the RegCM4-NH dynamical core is that, especially for long runs
with varied meteorological conditions, a relatively short time step is needed for stability
reasons. This makes the model rather computationally demanding, although not more
than other convection-permitting modeling systems such as the Weather Research and
Forecast model (WRF, Skamarok et al. 2008). For this reason, we are currently
incorporating within the RegCM system a very different and more computationally efficient
non-hydrostatic dynamical core, which will provide the basis for the next version of the
model, RegCM5, to be released in the future.

Following the philosophy of the RegCM modeling system, RegCM4-NH is intended to be
a public, free, open source community resource for external model users. The non-
hydrostatic dynamical core has been implemented in a way that it can be activated in
place of the hydrostatic dynamics through a user-set switch, which makes the use of
RegCM4-NH particularly simple and flexible. We therefore envision that the model will be
increasingly used by a broad community so that a better understanding can be achieved
of its behavior, advantages and limitations.

**Code availability**:  https://zenodo.org/record/4603556
**Cases study configuration files:** https://zenodo.org/record/5106399

**Author contribution**: CE prepared the manuscript with contributions from all co-authors
and coordinated research, SP, TA, GR carried out and analysed the simulations, PE
investigated solutions to stabilize/adapt the model at the km-scale and performed
preliminary validation tests, GG developed/adapted the model code, FDS contributed to
develop the coupled version of the model, NR developed one of the microphysics
scheme, GF supervised and coordinated all activities.

**Competing interests**: The authors declare that they have no conflict of interest.

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
