# Peer review of "Non-Hydrostatic RegCM4 (RegCM4-NH): Model description"

_Geoscientific Model Development, 2020_

## Author Comment (AC1)

**Non-Hydrostatic RegCM4 (RegCM4-NH): Model description and case studies over multiple domains**

Erika Coppola, Paolo Stocchi, Emanuela Pichelli, Jose Abraham Torres Alavez, Russell Glazer, Graziano Giuliani, Fabio Di Sante, Rita Nogherotto, and Filippo Giorgi

**Referee #1 (Citation**: https://doi.org/10.5194/gmd-2020-435-RC1**)**

This paper introduces the development of RegCM4-NH, and shows three simulation cases. It is an important paper introducing a new member to the convection-permitting simulations. The manuscript is well organized and easy to follow. Before the paper can be accepted, more details should be further clarified.

**Response**: Thanks to the Reviewer for the time she/he dedicated to review our manuscript. Below our responses to the comments.

Major comments:

1. Authors state that the stability is a quite important factor to be considered when the RegCM4 is switched to NH core. Ten major modifications are implemented comparing to the original MM5 code. The explanations on why each modification is added should be provided. For example, the proper choice on the schemes of 0nd diffusion and their combination is quite important for the computational stability. The dynamical core used here relies on explicit numerical diffusion to be numerically stable. How the advection term chosen here considering both the stability and accuracy should be introduced more clearly, and the proper references are needed here.

**Response**: We accepted this suggestion and we have added references to the base finite differences technique used, along with a description of the adopted discretization for the advection equation. Herein, we provide to the reviewer a plot showing the result in a 1D case for a discontinuous signal advected along the equator with a perturbed velocity on a staggered $X,U$ grid for both the original MM5 discretization and the RegCM local Courant number weighted discretization. After 48 hours of integration, the resulting advected signal shows a halving of the computational mode noise generated by the CTCS scheme, even when applying to both numerical schemes the same Robert-Asselin filtering. Because the implemented interpolation cannot be considered a novel numerical scheme but a slight modification to well-consolidated methods, the authors do not deem necessary for this paper an in depth analysis of the scheme stability or accuracy, which are nevertheless second-order accurate in the model discretization parameters. The change of the Laplacian stencil in the explicit diffusion to reduce the computational cost of communications is now described in the manuscript. The other changes listed in the manuscript allow the user explicit namelist control over model parameters that in the MM5 are reported as configurable in the code. The top radiative boundary condition filter coefficients are computed every 24 hours integration time to adapt it to the model internal solution for a climate simulation period longer than the weeklong ones typical of the MM5 NWP code.

[Figure]

**Figure R1**. 1D case for a discontinuous signal advected along the equator with a perturbed velocity on a staggered $X,U$ grid for both the original MM5 discretization (top panel) and the RegCM local Courant number weighted discretization (middle panel).

2. Different observation datasets are chosen for three cases. But in fact, both the CHIRPS and CMORPH can cover all three cases, and the NCEP data can cover two US cases. It is necessary to use the same observation references to evaluate the simulations. Suggest to show all the observation data considering the uncertainties. Or necessary explanations on such choice are needed.

**Response**: Following the Reviewer remark we now use several and common observational dataset available for each area as suggested also by Prein and Gobiet (2017) findings . Figure 4 has been splitted in three different figures (figures 5, 7 ,11) and we describe each event separately. We also have added information about the observed dataset in the manuscript and summary table (Table 3). The discussion about each case study has been revised accordingly (changes are tracked in the manuscript).

3. In the case LKV, the proper simulation on the contrast between land temperature and lake temperature is important on reproducing the local circulations. So figures on surface temperature from both 3-km and 12-km simulations are necessary to check whether the underestimation on rainfall from 12-km simulation is induced by the biases in surface temperature. And similar figures based on 12-km simulation should be added in the Figure 7.

**Response**: Following the Reviewer remark we prepared a Longitude-time (hourly) Hovmöller diagram (Figure 10 ) of LKV domain surface temperature in order to evaluate the difference in terms of temperature gradient between the two simulations (tracked in *Lake Victoria* section*)

Other comments:

1. The domains of 12-km simulations can be shown.

**Response**: done

2. The namelist files used for three cases should be included in the model codes, then the RegCM-NH can be easily used by the RegCM modeling community. And the choices on schemes of other physical processes should be introduced in the manuscript, such as the PBL.

**Response:** Namelist files are now available in **https://zenodo.org/record/5106399** and referenced in the manuscript.

3. L50: REGCM should be RegCM
4. L52: rcp should be RCP
5. Table 1: The year is missing in case 2
6. L319: Era should be ERA
7. It is hard to get the values from Figure 4 under the current color set.

**Response:** Unless differently indicated, the list above (3-7) has been fully considered and errors/typos/modifications have been implemented or revised in the manuscript.

---

## Author Comment (AC2)

**Non-Hydrostatic RegCM4 (RegCM4-NH): Model description and case studies over multiple domains**

Erika Coppola, Paolo Stocchi, Emanuela Pichelli, Jose Abraham Torres Alavez, Russell Glazer, Graziano Giuliani, Fabio Di Sante, Rita Nogherotto, and Filippo Giorgi

**Referee #2 (Citation**: https://doi.org/10.5194/gmd-2020-435-RC2**)**

The paper "Non-Hydrostatic RegCM4 (RegCM4-NH): Model description and case studies over multiple domains" describes the extension of RegCM4 with a non-hydrostatic option. Three case-studies are presented which all feature heavy precipitation events in different parts of the world.

**Response**: Thanks to the Reviewer for the time she/he dedicated to review our manuscript. Below our responses to the comments.

General comments:

The paper could make a valuable contribution to the community, but it seems unfinished and needs major improvements.

1. The current manuscript needs a general language check. One can often find slips like additional blank spaces or inconsistent naming. Just to give one example: The term convection-permitting is used often in the text but sometimes with and sometimes without hyphen. It feels like the text has been written by many different people, which is not a bad thing at all, but it adds to the impression of being unfinished. Some sort of harmonization by one author would increase readability and consistency.

**Response**: Thank you to the Reviewer for her/his comment. We have corrected typos and slips. We did a language check, rephrasing wherever possible to better harmonize the text.

2. Another aspect that makes the manuscript look unfinished is the fact that some features of the model are explained in great detail, but others are completely left out. The title reads "Model description". I do not expect that all components are described in great detail, but at least a table listing all model features such as radiation scheme etc. with references where to find a description would be nice. Not everyone knows RegCM4 and searching all the other references for the bits an pieces is quit cumbersome. What is the time integration scheme?

**Response**: We have added Table 1 with a full list of physics parameterizations and references. The time integration scheme used by the RegCM model is the leap-frog one. This information has been added to the manuscript.

Specific comments:

Line 24: Delete "the" in front of the first RegCM. What does RegCM stand for? **Response:** The sentence relative to the Regional Climate Modeling system (RegCM) here (L24-27) has been opportunely rephrased.

Line 50: Do you mean "bias compared to observations"? **Response:** correction done

Lines 65-67: The mentioning of Grell et al. (1995) seems redundant in this sentence. **Response:** correction done

Line 68: Do you mean "same grid an variable structure as RegCM4"? **Response:** rephrased

Line 229-231: For the Texas case you used ERA-Interim directly. Can you also motivate this decision with spatial spin-up and the work by Matte et al. (2017; DOI: 10.1007/s00382-016-3358-2)?

**Response:** We appreciate the comment and we added some discussion in the text. In the Texas case study we nested the model directly in the ERA-Interim reanalysis with boundary conditions provided every 6 hours, given that such configuration was able to reproduce accurately the HPE intensity. In this case the model uses a large LBC relaxation zone which allows the description of realistic fine-scale features driving this weather event (even if not fully consistent with the Matte et al., 2017, criteria).

Table 1: Can you add the domain sizes. **Response:** done. Table 2 (1 in previous version) now also includes domain size information.

Figure 2: Numbers and texts around the sub-figures are too small. Is it possible to use a common label-bar and the same contour intervals? This would make it easier to distinguish different orographic features in the domains. In section 3.1 you describe the Russian River. Maybe it is worth to already indicate it here, as not everyone is familiar with California.

**Response:** We improved figures readability. Moreover we have indicated the area where the russian river is located with a red dot in the new domains figure (Fig. 3) and we have added a short description in the text.

Figure 3: Numbers are too small. What does the arrow length mean? I find it hard to compare the maps a) and b) with c). Can you either choose the same section in c) or at least indicate the regions of a) and b) in c) with a square.

**Response:** The panels (now Figure 4) have been modified accordingly. Unit vector has been added to give the measure of wind velocity based on arrow length. The whole figure has been replotted to improve its readability. A black square in panel "c" surrounds the same area as in panels "a,b".

Line 271: Delete "the" in front of land. **Response:** The text has been rephrased accordingly to the new analysis.

Line 274: Do you mean "same variables from ERA5"? **Response:** yes. rephrased (now line 408).

Figure 4: Numbers are too small. I would recommend to split the figure and treat each region separately. All cases use different observations or analysis products and one case does not have a corresponding hydrostatic run. Keeping everything in one figure is not helping the reader, but tends to confuse. What are the resolutions of the observations/analysis?

**Response:** Done. We have splitted Figure 4 in three different figures (figures 5, 7 ,11) and analyzed each case separately using several observational datasets as also required by the another referee.

Line 319: Is the comparison to ERA-Interim really fair given that the jump in resolution is rather extreme? One could even argue that the precursors are in ERA-Interim, because the downscaling captures the event.

**Response:** Thank you for the comment. In Figure 7, we now also show five other high-resolution observational datasets for comparison. We included the ERA-Interim because it's useful to see how the driving boundary conditions represent the rainfall event and it highlights the advantages of the high-resolution simulation, but it was not our intention to make an unfair comparison. We have rephrased opportunely and also added a discussion about the other observational datasets.

Line 329: "The RegCM4-NH simulation shows a more realistic temporal evolution than the RegCM4, ..." Did you mean ERA-interim instead of RegCM4? I thought there was no (hydrostatic) RegCM4 simulation. **Response:** Right. We have corrected the sentence (now line 498).

Line 360: Can you motivate the choice of 26°C for the lake temperatures with observations?

**Response:** This initial LST value was chosen following some previous studies. Talling (1969) shows for the Lake Victoria surface temperatures ranging from 24.5-26°C during the course of the year. Several studies have used RCMs to investigate the climate over Lake Victoria (Anya et al., 2006; Anyah and Semazzi 2009, Sun et al. 2015). They showed a significant relationship between lake temperatures and rainfall which varied depending on season. The value of 26°C (Tailing 1969, YIN X. and Nicholson S.E. 1998) was chosen based on preliminary sensitivity tests using different values of temperature ranging from 24°C to 26°C in order to evaluate the effect of lake surface temperature on RegCM simulation and on lake-atmosphere coupling in the representation of the spatial distribution and iIntensity of the precipitation over the Lake Victoria Basin (Sun et al 2015). The initial LST of 26°C has shown to produce the most realistic precipitation for the period analyzed. We have added this discussion in the manuscript.

Figure 6: Numbers are too small. Coastlines are very hard to see. What does the arrow length mean? **Response:** Done, We have changed the figure (now Figure 8) in order to make it more clear/comprehensible. Unit vector has been added to give the measure of wind velocity based on arrow length.

Line 374: Can you indicate the cross-section with a line in Figure 6  **Response:** We have added the cross section in Figure 8b.

Line 393: Replace "and this" with "which". **Response:** Done. The whole sentence has been rephrased (L 619-621).

Lines 394-395: This sentence is hard do understand. Do you mean that the maximum is captured but the pattern shifted to the south? Please rewrite.

**Response:** The sentence has been rephrased: **"**In particular, the 3 km simulation reproduces well the local rainfall maxima on the western side of the lake, although these appear more localized and with a multi-cell structure compared to CMORPH and TRMM.**"**

Line 398: Delete "overall" **Response:** The whole sentence has been rephrased.

Line 399: Delete "that" **Response:** The sentence has been rephrased.

Section 4: The first paragraph needs a complete revision. I find many formulations hard to understand (e.g. the sentence from line 406-408). I can only find one conclusion that basically reads that non-hydrostatic models are better in simulating convection than coarser hydrostatic models. This is not new and obvious to me. I'm not saying that a model description paper needs ground braking conclusions. I rather like to encourage the authors to think about the possibilities the new system is opening up and how this system can add to the challnges around local climate change. The second paragraph is touching this topic and maybe the authors can expand on this. I would even wish for a short section in the main text on the performance of the model on climatic time scales.

**Response**: Sentence in lines 406-408 (now L 635-637) has been rephrased to clarify.
Findings from first long-term multi-model experiments at the convection permitting scale are discussed in the Introduction section documenting recent literature on the topic (Coppola et al., 2020, Ban et al., 2021, Pichelli et al., 2021). A second paragraph in the Conclusion section has been added.

For completeness we have adapted some plots from Pichelli et al. (2021) to show the behaviour of RegCM-NH in terms of climatic simulations within the km-scale multi-model ensemble for a series of precipitation indices. We report below the box plots, Fig. R2-1(Fig. R2-2), of area-averaged precipitation indices (mean precipitation, wet-day(hour) intensity, wet-day(hour) frequency, heavy precipitation (P99 for day extremes, P99.9 for hourly ones)) across the 12 models ensemble at km-scale resolution used in the paper, highlighting with a black cross the position of RegCM-NH within the ensemble. The models common area of study and the sub-regions considered are reported in Fig. R2-3. The ensemble and RegCM-NH are compared with different observed dataset at the daily scale (Figure R2-1) and with the highest (space-time) resolution observations at the hourly scale (Figure R2-2). The model

performance changes on a season and region base, showing a greater tendency to be in line with the rest of the ensemble (first and third quartile) in fall season, while often laying at the drier edge of the distribution in summer, although remaining within the 5th percentile.

[Figure]

**Figure R2-1** Readapted from Pichelli et al. (2021) JJA (left) and SON (right) red box-plot representing the distribution of the convection-permitting models of their ensemble over different areas (CA: common domain as represented in Fig. R2-3 below, CH: Switzerland, (N/C/S)IT:(North/Central/South) Italy, (S)FR: (South) France) over the historical period 1996-2005. The distribution is represented within 5th and 95 percentile, for (from top to bottom) mean daily precipitation (mm/day), mean wet-day intensity (mm/day), wet-day frequency, heavy daily precipitation (p99, mm/day). RegCM-NH is represented by the black cross; outliers are represented singularly (open diamonds), ensemble mean (yellow line) is also reported and compared with the highest resolution observations (white star, RdisaggH (CH), COMEPHORE (FR), GRIPHO (IT)) over the same area and other dataset at coarser resolution where available (EURO4M-APGD (yellow pentagon), E_OBS 25 km (pink pentagon), SAFRAN reanalysis 8km (green pentagon).

[Figure]

**Figure R2-2** Readapted from Pichelli et al. (2021) Same as Figure R2-1 but for (from top to bottom) mean wet-hour intensity (mm/h), wet-hour frequency, heavy hourly precipitation

(p99.9, mm/h). Only the highest resolution observed dataset is reported here (star, RdisaggH (CH), COMEPHORE (FR), GRIPHO (IT)).

[Figure]

**Figure R2-3** by Pichelli et al. (2021): Common domain (CA). Areas where observed dataset are available are in different colours: EURO4M-APGD (dashed grey), REGNIE (yellow), Spain02 (pink), RdisaggH (red), COMEPHORE (blue), GRIPHO (greenish). Sub-domain areas considered in the analysis are labelled: South France (SFR), North, Central, South Italy (N/C/SIT), Switzerland (CH) ( Pichelli et al, 2021, their section 3 for dataset description).

---

## Author Response (AR1)

[revised manuscript text omitted]

---

## Author Response (AR2)

**Non-Hydrostatic RegCM4 (RegCM4-NH): Model description and case studies over multiple domains**

Erika Coppola, Paolo Stocchi, Emanuela Pichelli, Jose Abraham Torres Alavez, Russell Glazer, Graziano Giuliani, Fabio Di Sante, Rita Nogherotto, and Filippo Giorgi

We thank the Reviewers for the time dedicated to the second review of the manuscript. We have revised the whole manuscript following their useful comments, which are point-by-point addressed below.

**Referee #1**

The revision really improves the quality of this paper, however a carefully language check is still needed, for example, the "Era-Int" or "ERA-Interim". Some minor revisions are needed, especially on

figures.

1. Figure 9: The high-resolution simulations always show strong vertical motion and accompanied convergence. It may be more proper to use different scale for 3-km and 12-km simulations, then the vertical and horizontal motions in the 12-km simulation can also be clearly shown. Shading not contours make it more aesthetic?

**Response:** We have tried using shaded contours instead but this makes it difficult to see the arrows. The arrows have been made bolder now along with the contour lines to improve the readability. About the horizontal scale, we believe it is important to keep a consistent scale between the 3km and 12km simulations so that it is easy to see the difference in the circulation in both. One of the points we make in this section is that the new 3km simulation has a stronger circulation response to convection, so it is important to show this comparison in Fig. 9.

2. Figure 10: zonal anomalies may help to highlight the lake-land differences.

**Response**: Done. Temperature anomaly are now displayed that better highlight the day-night and the lake-land contrast

3. Figure 3 caption: L369 "(c)" should be "(b)" **Response**: Right. Corrected.

4. Figure 6 caption: "by the red square in Figure 3 (a,b)" ? **Response**: This has been corrected. The black squares in figures 5a and 7a.

5. All the ocean and lake coastlines should be thick enough

**Response**: Done. Lake/Ocean coastlines have been plotted thicker to improve figures readability

##############################################################################

**Referee #2**

This is the 2nd round of review for this article.

General comments:

- The manuscript improved a lot since the first round, but there are still a couple of language issues (some examples will be given later). I think the authors could improve by formulating shorter sentences. Many hard to understand formulations come from long sentences with many subclauses. Especially in the section about Lake Victoria this becomes an issue.

**Response**: Done. Text has been revised to improve readability.

- Some figures still need improvements in terms of readability. Many contain latitude/longitude information that are too small to read.

**Response**: Most of figures have been revised for improving readability

- Please double check how dates and times are written. For dates it should be either June 12 or 12th of June, but not 12 June. Is the time of the day in all cases local time? For Lake Victoria you explicitly state 12Z (better would be 12:00Z), which I would refer to UTC. Later in that section (Figure 10) you write something like "25Nov h:06". Is this UTC, too? I don't want to be too picky here, but at least it should be consistent throughout the paper.

**Response**: the dates and time are uniformly written now.

- Please add degree signs to latitude/longitude information.

**Response:** Done. Coordinate labels are shown with West (W) and East (E) lettering.

- Figure 9+10: The section around Figure 9 and Figure 10 (L561-L580) is very ambiguous to me. This needs much better explaination including the captions. Some unclear points are detailed below.

**Response:** Text has been revised and in particular Figure 9 captions improved. Figure 10 has been replaced as suggested by reviewer #1, now showing more clearly lake/land and day/night gradients as reproduced at the two resolutions through surface temperature zonal anomaly.

Specific comments:

L342: is not there -> is lacking **Response**: Done

Figure 3: lat/lon information too small. The caption needs to be rephrased. **Response**: Done

L376: Figure 3 -> Figure 3a **Response**: Done

L387: upslope flow Raplh et al. (2006) -> upslope flow (Raplh et al., 2006) **Response**: Done

Figure 4: The units for mslp and surface temperature are missing. Numbers at the colorbar are too small. Arrow legend is too small. **Response**: units for mslp and surface temperature have been added in the caption. We have increased the size of the colorbar numbers and of the arrow legend (as well)

L429: delete blank after precipitation. **Response**: Done

Figure 5: Please add letters to the sub-figures and update the text accordingly. lat/lon information too small. **Response** : Done. We have labeled panels and improved plot readability.

Figure 6: Please add latters to the sub-figures and update the text accordingly. **Response**: The letters are already indicated in the previous version

Figure 7: Please add letters to the sub-figures and update the text accordingly. lat/lon information too small. **Response**: Done. We have labeled panels and improved plot readability.

L552: Do you mean Figure 8a? **Response**: Right. Corrected.

Figure 8: The units for mslp and surface temperature are missing. Numbers at the colorbar are too small. Arrow legend is too small. **Response**: units for mslp and surface temperature have been added in the caption. We have increased the size of the colorbar numbers and of the arrow legend (as well)

L564: right panel of Fig. 8 -> Fig. 8b) **Response**: Corrected.

L566: differential -> difference **Response**: Corrected.

Figure 9: Please add unit for the y-axis. The caption needs a rework. See details in following comments. **Response:** Done**.** The figure caption has been revised pressure/units label added to the y-axis. Some figure aspects have been revised to improve the readability .

L583: Where do I see a red line in Fig. 9? Do you mean the black line in Figure 8b)? **Response**: Right. Corrected.

L584: Which zonal-wind anomaly is meant here? This is not explained anywhere else. Why is the mean of 0-2N (I assume 0°-2°N) shown when the cross-section is at 1°S? **Response:** Caption has been revised and information on the zonal-wind anomaly added to the section which introduces this figure. The zonal-wind anomaly is (0°-2°S) and was incorrect in the caption. Corrected now.

Figure 11: Please add letters to the sub-figures and update the text accordingly. lat/lon information too small. What are the contours? Which unit? **Response**: Done. Units for precipitation have been added in the caption.

L611: climatology -> climatologies. **Response**: Done

L617: Please rephrase that sentence. **Response:** Done. We rephrased to clarify. "However, despite the large uncertainty among the different observed datasets (Figure 11 a-d), we find a significant underestimation of the precipitation by the 12 km run over the lake independently of the dataset used as a reference (Figure 11f)."

L634-637: This sentence does not make sense. **Response:** We rephrased and corrected the typo of MM5 instead of MM4 hydrostatic core. "The non-hydrostatic dynamical core of MM5 has been thus incorporated into the RegCM4 system previously based on the MM5 hydrostatic core."

L651: convection permitting -> convection-permitting **Response**: Done

L657-659: What do you mean with short time scales? Do you refer to events of short duration, e.g. sub-daily heavy precipitation? With this I could agree. Maybe you can rephrase this sentence. **Response:** Right, we intended sub-daily instead of short. This has been corrected in the text.

---

## Author Response (AR3)

Dear Editor,

we do regret for the error. We would have never revised the manuscript presuming this would not be checked because is also our interest to improve the manuscript following Reviewer comments as we are used to do in all the research papers we publish. In case we do not agree with the Review comments we point this out in the report highlighting which suggested modification was not accomplished (motivating our choice eventually).

Just to explain what happened, we have worked on a google doc file that has been then downloaded before sending back to the Journal.
Most of the revised figures were inserted in "suggestion mode" (to track the change). Differently to what happened with changes in the text, once downloaded the manuscript offline, most figures inserted this way were not saved and we did not realize that until your email.

We are sorry and we are providing the revised version of all figures in the manuscript, including the list of missing changes that you have highlighted.

Best regards

Erika Coppola on behalf of all coauthors

The full list below has been considered.
* Fig. 3: I do not see, that the lat/lon information increase
* Fig.4 The numbers on the color bar are no longer complete
* Fig. 5 + 7: I do not see letters on the subfigures / visibility of lon/lat labels did not improve
* Fig. 8: size of number in colorbar did no change, size of arrow legend is the same
* Fig. 11: labels a)-f) in panels are missing / size of lat/lon did not change

* unit Kelvin [K] is without degree, therefore remove it from all panels and captions.

---

## Author Response (AR4)

Dear Editor,

thank You for your feedbacks. We have finally corrected also the colour bar in Figure 4.

Best regards

Erika Coppola on behalf of all coauthors